# Can Blood Flow Restriction Training Benefit Post-Activation Potentiation? A Systematic Review of Controlled Trials

**DOI:** 10.3390/ijerph191911954

**Published:** 2022-09-21

**Authors:** Haodong Tian, Hansen Li, Haowei Liu, Li Huang, Zhenhuan Wang, Siyuan Feng, Li Peng

**Affiliations:** 1Key Laboratory of Physical Fitness Evaluation and Motor Function Monitoring, Southwest University, Chongqing 400715, China; 2Department of Physical Education, Southwest University, Chongqing 400715, China; 3Institute for Health and Sport (iHeS), Victoria University, Melbourne, VIC 3011, Australia; 4Laboratory of Genetics, University of Wisconsin—Madison, Madison, WI 53706, USA

**Keywords:** post-activation potentiation, blood flow restriction, conditioning activities, athletic performance, vertical jump, bench press

## Abstract

(1) Background: post-activation potentiation (PAP) plays an essential role in enhancing athletic performance. Various conditioning activities (CAs) have been developed to generate PAP before training or competitions. However, whether extra equipment can enhance the effectiveness of CAs is understudied. Hence, this systematic review aims to introduce and examine the effectiveness of blood flow restriction-based conditioning activities (BFR-CAs). (2) Methods: a literature search was conducted via Web of Science, PubMed, SPORTDiscus, and CNKI (a Chinese academic database). The systematic review included the literature concerning BFR-CAs and non-BFR-CAs. The methodological quality of included studies was considered to be “moderate quality” and “good quality” based on the Physiotherapy Evidence Database Scale. (3) Results: five studies were included in this study. Four studies were on lower limb strength training, and three of them suggested a greater PAP in BFR-CAs than in non-BFR counterparts. One study on upper limb strength training also supported the advantage of BFR-CAs. (4) Conclusions: BFR-CAs may be an emerging and promising strategy to generate PAP. Compared with non-BFR-CAs, BFR-CAs might be more efficient and practical for inexperienced sports people or athletes in non-power sports.

## 1. Introduction

Strength/power production plays a critical role in athletic performance, including sprinting, jumping, throwing, kicking, and changing direction [1]. Hence, strategies to enhance strength/power production draw considerable attention from trainers and trainees. Currently, there is a general agreement that a maximal or near-maximal muscular contraction could increase strength/power production in subsequent exercises. Post-activation potentiation (PAP) is defined as such an increase and has inspired many new training strategies. Theoretically, PAP refers to the temporary increase in strength and the rate of force development caused by pre-activation [2]. Since PAP can contribute to a high level of enhancement in athletic performance [3], various conditioning activities (CAs) have been developed to generate PAP in order to optimize athletic performance, including improving explosive power [4], endurance [5] and specific sports technologies [6]. In addition to athletic training, CAs are also advocated in the rehabilitation of athletes. Empirical evidence suggests that high-load training is essential to maximize PAP [7,8]. This is because high-load training is more conducive to muscle activation [9], which is believed to be a physiological mechanism behind PAP [10]. Therefore, existing CA protocols usually use training loads between 80% and 100% of 1 repetition maximum (1RM) to induce PAP [11]. However, traditional high-load training can easily cause intensive fatigue. MacIntosh et al. [12] suggest that fatigue can limit subsequent potentiation, thereby decreasing athletic performance. In addition, extremely high-load training can cause damage to muscle microstructure, which can increase the risk of injury [13]. These issues may mask the benefits of CAs and even limit the long-term development of athletes. Furthermore, individual characteristics (e.g., strength, skill level, proportions of fast-twitch muscle fibers, etc.) may cause different adaptability to high-load CAs among trainees [14].

In order to maintain the benefits of CAs and reduce the risks of high-load training mentioned above, it has been recommended that extra training equipment is incorporated with CAs. Blood flow restriction (BFR) is a training method that partially restricts arterial inflow and fully restricts venous outflow in working musculature during exercise [15]. BFR can be applied during resistance training, aerobic training, and even non-traditional exercise such as whole-body vibration training [16], which generally aims to promote muscle hypertrophy and strength. In addition, low external load demand in BFR training allows it to benefit both healthy people and load compromised populations in need of rehabilitation [17,18]. Furthermore, more recent research has documented that BFR training may increase muscle activation both in sports training [19,20] and sports rehabilitation [21,22]. Therefore, some recent studies have applied the BFR method to CAs (BFR-CAs) and found enhanced athletic performance as a result of greater PAP [6,23]. That means BFR-CAs may be substitutes for traditional CAs that can easily cause injury and over fatigue. However, due to limited sample sizes and some controversial results, the effectiveness of BFR-CAs is not very clear. In response, the current study was conducted to systematically review the experimental studies concerning BFR-CAs and offer qualitative evidence to support the advantages of BFR-CAs over non-BFR-CAs.

## 2. Methods

### 2.1. Data Sources and Search Strategy

A systematic search was conducted via databases including Web of Science, PubMed, SPORTDiscus, and CNKI (a Chinese academic database). The time frame for the search was between the dates the databases were set up and June 2022. The search was performed using the following terms alone or in combination: (“PAP” [Title/Abstract] OR “post-activation potentiation” [Title/Abstract] OR “post-activation potentiation” [Title/Abstract] OR “higher motor unit activation” [Title/Abstract] OR “muscle excitation” [Title/Abstract]) AND (“blood flow restriction training” [Title/Abstract] OR “BFR training” [Title/Abstract] OR “KAATSU training” [Title/Abstract]). Additionally, we used a back-and-forth strategy to identify potential studies via the references of primarily included studies. For any unavailable papers, we tried to contact the authors for the full text.

### 2.2. Eligibility Criteria

All the studies with full text were screened independently by two reviewers (HDT and HWL) according to the PICOS criteria [24,25]:P (Participant): Involved subjects with no known medical conditions or injury.I (Intervention): BFR-CAs with clear load.C (Comparison): non-BFR-CAs with clear load.O (Outcome): Any validated measure of PAP (e.g., lower-limbs explosive performance via vertical jump height, flight time, power, or electromyography (EMG) of the vastus lateralis and hamstrings; upper-limbs explosive performance via the bar velocity, power output of bench press), assessed using PAP indicators at both pre- and post-intervention. There must be a completely negative rest interval between training and PAP test.S (Study design): Controlled trials, with RCT given priority.

### 2.3. Data Extraction

For all included articles, the following data were extracted: (a) study characteristics (author, year, and sample size); (b) subject demographics (sex, exercise experience, and allocation); (c) CA protocols (exercise, load, pressure of BFR, and interval); (d) rest interval (period between training and test); (e) study results (changes of indicators concerning PAP); (f) study conclusion. Then, data for the pre- and post-training means and standard deviations of the included were coded. Due to the limited number of included trials and the inconsistent training loads, quantitative analysis was deemed inapplicable to this study.

### 2.4. Methodological Quality Evaluation

The modified Physiotherapy Evidence Database (PEDro) scale was employed by 2 independent investigators to evaluate the methodological quality of the studies included in the review, and mutual agreement was obtained for any observed discrepancies. The original 11-item PEDro scale involves eligibility criteria (item 1), randomization (item 2), concealed allocation (item 3), similar baseline (item 4), blinding of all participants (item 5), blinding of all therapists (item 6), blinding of all assessors (item 7), more than 85% retention (item 8), intention-to-treat analysis (item 9), between-group comparison (item 10), and point measures and measures of variability (item 11). Since it is not generally feasible to blind the subjects and investigators in supervised exercise interventions, we removed items 5–7 from the scale, which are specific to blinding. This approach has been used in previous systematic reviews in the area of exercise [26]. After removing these items, the maximum result on the modified PEDro scale was 7 because the first item (which relates to the eligibility criteria) is not included in the total score. The studies were categorized as follows: 6–7 = “excellent quality”; 5 = “good quality”; 4 = “moderate quality”; 0–3 = “poor quality”. This is consistent with previous exercise intervention reviews [27,28].

## 3. Results

### 3.1. The Results of Literature Retrieval

A total of 132 articles were initially identified, with 32 from the web of science, 93 from PubMed, 4 from SPORTDiscus, and 3 from CNKI. Then, 40 duplicates were removed, and 81 articles were excluded based on their titles and abstracts. After reviewing the full text, six articles were included for the review. Among the included studies, one tested peak power output, mean power output, peak bar velocity, and mean bar velocity to detect PAP [23]. The remaining four articles tested vertical jump [6,29,30,31,32]. The flow diagram is presented in Figure 1.

### 3.2. Methodological Quality

The PEDro scores for the studies in this review ranged from 4 to 6 (mean = 5.2 ± 0.8) (Table 1). Of the five studies, two had a total score of 6, two had a total score of 5, and one had a total score of 4. These results indicate that the evidence used in this review comes from studies with “moderate quality” to “good quality” methodological quality.

### 3.3. Characteristics of Studies Included

A total of five studies with 81 subjects were included in this systematic review (Table 2). Their study areas were the United States, Australia, the United Kingdom, Poland, and China, and their sample sizes were between 20 and 10. They were published between 2017 and 2022. Their outcome indices included bench press power, bar speed, EMG index, vertical jump height, vertical jump power, flight time, and other indicators. Among the included studies, one study focused on the gain of upper limbs’ strength performance [23], and four studies focused on the improvement of lower limbs’ strength performance [6,29,31,32].

## 4. Discussion

### 4.1. Magnitude of PAP Elicited by BFR-CAs

The main purposes of this study were to examine the role of BFR in eliciting PAP and to compare the effectiveness of BFR-elicited PAP and non-BFR counterparts. Given the limited number of included trials and variable training loads, quantitative analysis was rejected. Nonetheless, most of the identified studies suggested the advantages of BFR training in eliciting PAP over non-BFR training. In all included studies, only one study documented adverse effects of both BFR and non-BFR-CAs on vertical jump performance, and the BFR even led to lower PAP. Likewise, null or negative results were reported elsewhere, despite those studies using resistance training alone without BFR for eliciting PAP [33,34,35]. Theoretically, the enhancement of athletic performance is decided by the net potentiate state resulting from the coexistence of fatigue and PAP [3]. Thus, over-fatigue may mask the effects of PAP and lower subsequent athletic performances. Generally, fatigue can be modeled by the metabolic pressure generated from specific metabolites. In addition, there is evidence supporting the conclusion that BFR training may cause more accumulation of inorganic phosphate and the decrease of pH in local muscle, which may compromise the contractile capacity of skeletal muscle through metabolic stimulation of group III and IV afferents (mechanoreceptors and metaboreceptors, respectively) and, consequently, reduced motoneuron activity (central mechanism) [36]. However, none of those studies measured fatigue in CAs. With regard to the study of Cleary and Cook [29], a 4 min rest interval may be insufficient for performance enhancement. In essence, there are two window times (one right after the CAs and the other one in the recovery period) allowing net potentiation to emerge. However, in this study, 4 min may lie just between the two window times. In other included studies, either multiple rest intervals or longer rest interval were chosen. To accurately check performance enhancement, it is suggested that in subsequent research, different rest intervals are chosen and changes of fatigue detected after CAs through biochemical detection.

Moreover, Chen et al. [37] observed an individual phenomenon of PAP in the response to various kinds of CAs. That is, for non-responders, there may be no window times and performance enhancement after CAs. While in all included studies, performance enhancement is defined by the average changes after intervention and little attention has been paid to subjects’ responses to CAs, it remains unclear whether the negative results are due to the rest interval, rather than the non-responders. Therefore, it is suggested that in future research, individual response to CAs is checked in advance.

One included study indicated negative results. The others demonstrated that BFR-CAs managed to elicit a better PAP. In particular, when LL-CAs were put into practice, BFR turned out to be more beneficial. In the study of Doma et al. [6] in which body-weight lunge exercise was used as the basic CA, BFR significantly enhanced the vertical jump height (~4.5% ± 0.8%), flight time (~3.4% ± 0.3%) and power (~4.1% ± 0.3%) within 6~15 min (*p* < 0.05), while no significant changes were elicited by its counterpart. In another study performing LL-CAs [32], BFR-CAs showed advantages over non-BFR-CAs in countermovement jump height, squat jump height, and reactive strength index (flight time/contact time). In addition, there was only one study included that used isometric training (isometric deadlift with most effort–MIVC deadlift). However, the use of non-BFR MIVC deadlift failed to significantly enhance vertical jump height and power in this study, despite other evidence supporting isometric contraction, especially MIVC, being a more favorable strategy for PAP elicitation [38,39,40]. However, after BFR MIVC deadlift, jump height appeared to significantly increase (*p* < 0.01). Apparently, the use of BFR boosted the elicitation of PAP. The results of these studies highlight the potential of BFR to enhance stretch-shortening cycle mechanics. They also indicate that there may be differences in time effects between BFR- and non-BFR-CAs.

### 4.2. Time Characteristics of PAP Elicited by BFR-CAs

Only one study reported the upper limb’s PAP elicited by BFR training. Wilk et al. [23] examined the PAP by recording the changes of power (peak power and average power) and bar speed (peak speed and average speed) during three sets of bench press training. Their subjects were ten males who were experienced in resistance training (with an average personal bench press record of 168.5 ± 26.4 kg). The results showed that the power and bar speed of the BFR group was significantly higher than the non-BFR group (peak power: 965 vs. 792 W; mean power: 667 vs. 559 W; peak speed: 0.74 vs. 0.62 m/s; mean speed: 0.53 vs. 0.46 m/s. *p* < 0.01). In addition, the performances of both groups significantly increased from set 1 to set 2, and the peak power and bar speed of those in the BFR group (*p* < 0.01) were significantly higher than the peak power and bar speed of those in the non-BFR group (*p* = 0.01), yet only the performance of the BFR group decreased significantly (*p* < 0.01) from set 2 to set 3, which suggested that BFR-CAs may elicit PAP faster than non-BFR-CAs. Similar results were reported in the study of Doma et al. [6] (BFR-lunge exercise enhanced vertical jump performances significantly after 6 min, while the enhancement did not appear until 15 min after non-BFR lunge exercise). However, the study of Wei and Xiang [32] found that PAP elicited by BFR-CAs existed longer that its counterpart (8 min vs. 4 min) which conflicts with the result of Wilk et al. [23]. This may be due to the different resistance load choice (70% 1RM and body weight). Therefore, in order to clarify the time effect differences., it is suggested that further multiple-time-point studies on different training loads with BFR are conducted.

Notwithstanding the limitations, it is proposed that BFR is conducive to the elicitation of PAP in its magnitude and speed. This result may partially support a previous study, where BFR triggered a greater neuromuscular adaption than non-BFR training [41]. Some clues or theories in the published literature may help to explain our findings. On the one hand, BFR training is more beneficial for muscle activation. Specifically, the hypoxia condition and the transfer of myofibrillar fluid caused by BFR lead to the premature fatigue of slow-twitch muscle fibers and, in turn, promote the recruitment of fast-twitch fibers [15,42,43]. Similarly, Yu et al. [44] have highlighted that BFR training can improve the coordination and regulation ability of the nervous system and therefore excite the central nerve and activate fast-twitch fibers. On the other hand, low load BFR training can significantly reduce large-scale muscle injury and severely delay muscle soreness (DOMS) while maintaining and even increasing training progress.

### 4.3. Individual Adaptability for BFR-CAs

Studies have suggested that personal characteristics, including training status, strength, and skill level, can positively affect PAP [45]. Specifically, PAP is positively associated with the proportion of fast-twitch fibers [3,46]. Untrained individuals or recreational trainees may not effectively recruit their fast-twitch fibers, so the PAP can be subtle or ineffective [47,48]. The current study suggests that BFR may serve both experienced and inexperienced athletes due to its adaptability. Wilk et al. [23] recruited well-trained bench press athletes (average 1RM = 168.5 ± 26.4 kg) and observed an increased power output and bar velocity of bench press after two sets of BFR bench press training. Doma et al. [6] found that after performing lunge exercises with BFR, anaerobically trained participants’ vertical jump height increased significantly after a 6 min interval, yet non-BFR lunge exercises did not show PAP until 15 min. Furthermore, although 8~12 min has been suggested as the optimal rest interval by others in traditional high-load training [49,50], the studies included here suggest using BFR-CAs eliciting PAP with rest intervals of between 6 and 10 min [30,31,32,51]. These findings imply that BFR-CAs may serve experienced athletes and may be more efficient than traditional CAs. Moreover, BFR-CAs may also help non-athletes because BFR may exert extra physical and physiological stress and, in turn, reduce the training loads required in CAs. For example, one study recruited college females and observed enhanced sprint performance after a BFR-CA [51]. Similarly, Miller et al. [31] deployed BFR-CAs in recreationally active men and observed enhanced jump performance. These clues may, to a certain extent, imply the effectiveness of BFR-CAs in non-athletes.

Taken together, relevant studies collectively suggest the role of BFR-CAs in eliciting PAP in non-athletes and athletes. For non-athletes, BFR helps them to share the benefits of CAs with lower technical requirements. For athletes, although traditional CAs are enough to induce PAP, BFR can be incorporated into their CAs to save energy for formal training or competition and also reduce the risks of injury in high-load training. However, current results may be unilateral, for most of the studies included recruited young individuals (18~35 years old). Most individuals, especially experienced trainees of these ages, are in good training condition. More experimental evidence on the effect of BFR-CAs on older or younger individuals (younger than 18) is needed.

### 4.4. Equipment Tips for BFR-CAs

Determining pressure modality in advance is essential for BFR training. The amount of pressure needed to cease blood flow to a limb is related to a range of individual limb characteristics (tourniquet shape, width and length, the size of the limb, or an individual’s blood pressure) and cuff width [16]. However, none of the included studies have elaborated the limb characteristics of subjects. It is proposed that the adjustment of pressure modality in protocols should be determined according to these individual parameters.

In the studies included here, most deployed a moderate cuff width of 6 cm, yet the pressures used were inconsistent. Miller et al. [31] and Wei and Xiang [32] used arbitrary and universal pressures (160 mmHg and 12.28 kPa) in their protocols, but due to individual differences, the blood flow restriction on each subject may be different. Although the absolute pressures in these two studies successfully potentiated sport performance, inconsistent training conditions may have affected the PAP elicitation of subjects in the same group. In the study by Doma et al. [6], pressure relative to systolic blood pressure was applied. Despite more individualization, inconsistent blood flow restriction may still have existed for the different cuffs used for traditional blood pressure and exercise. Moreover, with regard to this study, whether the SBP measured from arms applies well for legs (larger limb) requires further examination. To individualize the blood flow restriction more directly, it is recommended that pressure is set during BFR-CAs based on arterial occlusion pressure (AOP).

## 5. Limitation

Admittedly, the present study is limited by the small sample size and the inconsistent loads of the CAs in the studies included. The small sample size is due to the limited research available on this topic, even though a systematic search via three global databases and the most authoritative database in China was performed. In addition, it was found that the main subjects of the studies included are resistance trained men. The limited studies and the biased sampling profile may limit the generalizability of the findings. Nonetheless, the work in this study indicates BFR-CAs may be used to elicit PAP in inexperienced individuals. Furthermore, the inconsistent resistance loads of the CAs in the studies included highlights the fact that the substitution relationship between HL and LL-BFR remains ambiguous, for the only research which compared these two loads suggested negative results. Therefore, it is suggested that future studies should be conducted that systematically examine the effect of BFR-CAs on inexperienced or non-power-sports athletes and explore the substitution relationship between HL- and LL-BFR-CAs and their effect on PAP. Furthermore, if and how the BFR affects the time characteristics of PAP also remains to be explored.

## 6. Conclusions

This review aimed to introduce BFR-CAs and compare their effectiveness in eliciting PAP with traditional CAs. Our results pointed to the advantage of BFR-CAs in inducing PAP. However, due to the limited number of relevant studies and the discrepancies in data synthesis, further research is warranted to re-examine the advantage of BFR-CAs when more studies appear. Moreover, since BFR itself can exert physiological stress to help induce PAP, BFR-CAs may require lower training loads to benefit untrained individuals and help save energy, and to reduce the risks of over-fatigue and injuries in athletes. This review thus highlights the possible value of BFR-CAs and the need to consider them further in future research.

## Figures and Tables

**Figure 1 ijerph-19-11954-f001:**
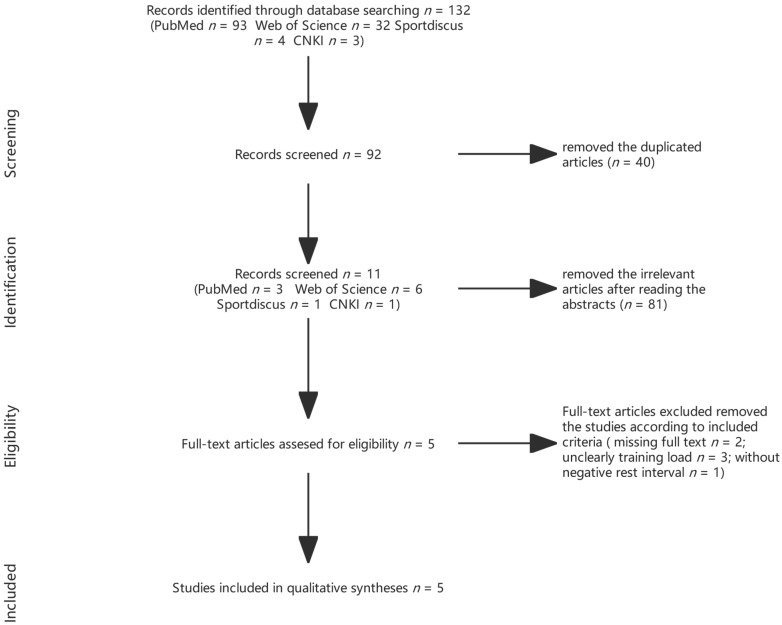
Schematic diagram of the article inclusion process.

**Table 1 ijerph-19-11954-t001:** PEDro ratings of the included studies.

Studies	Criteria								
	1	2	3	4	8	9	10	11	Total
Wilk et al. [23] (2020)	Yes	0	0	0	1	1	1	1	4
Cleary and Cook [29] (2020)	Yes	1	0	1	1	1	1	1	6
Doma et al. [6] (2019)	Yes	0	0	1	1	1	1	1	5
Miller et al. [31] (2018)	Yes	0	0	1	1	1	1	1	5
Wei and Xiang [32] (2022)	Yes	1	0	1	1	1	1	1	6

Items in the PEDro scale: 1 = eligibility criteria were specified; 2 = subjects were randomly allocated to groups; 3 = allocation was concealed; 4 = the groups were similar at baseline regarding the most important prognostic indicators; 8 = measures of one key outcome were obtained from 85% of subjects initially allocated to groups; 9 = all subjects for whom outcome measures were available received the treatment or control condition as allocated or, where this was not the case, data for at least 1 key outcome were analyzed by “intention to treat”; 10 = the results of between-group statistical comparisons were reported for at least one key outcome; 11 = the study provided both point measures and measures of variability for at least one key outcome. 1 = explicitly described and present in details; 0 = absent, inadequately described, or unclear.

**Table 2 ijerph-19-11954-t002:** The characteristics of the included studies.

Studies	Subjects	Protocols	*N*	Pressure Modality	Exercise	Reps/Sets	Interval	Rest Interval	Main Results	Conclusion
Wilk et al. [23] (2020)	Resistance trained men (age: 29.8 ± 4.6 years; body mass:94.3 ± 13.6 kg; BP 1RM: 168.5 ± 26.4 kg)	HL(70% 1RM)\HL(70% 1RM) + BFR	5\5	90%AOP, Cuff width: 6 cm	Bench press	Both 3/3	5 min	Real time recording	Peak power: compared with HL group, HL + BFR group increased significantly in the second group (*p* < 0.01) but decreased significantly in the third group (*p* < 0.01). Peak bar speed: the change is the same as the peak power. Average power and average bar speed: HL + BFR group was significantly higher than HL group in each period (*p* < 0.01).	BFR training can enhance strength performance significantly and is suitable for experienced trainers.
Cleary and Cook [29] (2020)	Resistance trained men (18~23 years old)	HL(85% 1RM)\LL(30% 1RM) + BFR	15\15	60%AOP, Cuff width: 6 cm	Squat	HL:5/2, LL + BFR:30/2	3 min	4 min	The incidence of PAP in the two groups significantly reduced. LL + BFR group: 90.8% ± 7.8%; HL group: 96.1% ± 7.8%.	Both BFR-and non-BFR-protocols weaken the subjects’ vertical jump performance
Doma et al. [6] (2019)	anaerobically trained men	LL (self weight)\LL (self weight) +BFR	9\9	130%SBP, Cuff width: not mentioned	Lunge	Both:8/3	2 min	3/6/9/12/15 min	LL + BFR group: jump height (~4.5% ± 0.8%), FT (~3.4% ± 0.3%) and power (~4.1% ± 0.3%) were significantly improved within 6–15 min post-exercise (*p* < 0.05); LL group: no significant changes (*p* > 0.05).	The BFR lunge squat improves the subsequent jumping performance of men undergoing resistance training. The use of BFR may be a practical alternative to HL training.
Miller et al. [31] (2018)	Recreationally active men (21.8 ± 6 2.6 Years old; 180.5 ± 6 6.2 cm; 84.5 ± 12.1 kg)	HL (most effort)\HL(most effort) + BFR	20\20	160 mmHg, Cuff width: 6 cm	Deadlift	Both 10 s/3	1 min	10 min	HL + BFR group: VJ-H 57.7 ± 7.9 and 59.4 ± 8.1 cm ↑ no significant increase in power; HL group: 59.7 ± 7.4 ~ 60.2 ± 8.6 cm ↑ (*p* < 0.01) no significant increase in power.	BFR training can enhance vertical jump performance significantly.
Wei and Xiang [32] (2022)	Resistance trained college students	LL(plyometric)\LL + BFR	9/9	21.28 KPa, Cuff width: 6 cm	Plyome-tric training	10/2, 5/3, 5/1	30 s/30 s/10 s	4/8/12/16 min	4 min: CMJ-H, CMJ-RFD, SJ-J, SJ-P, RSI significantly increased in both groups (*p* < 0.05). 8 min: CMJ-H, SJ-H and RSI significantly increased in LL-BFR group.	Both plyometric training and BFR + plyometric training can significantly enhance PAP. And PAP induced by BFR + plyometric extended to 8 min.

HL, high-load; LL, low-load; RM, repetition maximum; MVC, most voluntary contraction; SBP, systolic blood pressure. Rest interval refers to the period between CAs and PAP test. VJ-H, vertical jump height; VJ-P, vertical jump power; DJ-H, drop jump height; DJ-P, drop jump power; CMJ-H, countermovement jump height; CMJ-P, countermovement jump power; SJ-H, squat jump height; SJ-P, squat jump power; RFD, rate of force development; RSI, reactive strength index; FT, flight time.

## Data Availability

Not applicable.

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
