# Peer review of "Can Blood Flow Restriction Training Benefit Post-Activation Potentiation? A Systematic Review of Controlled Trials"

_ijerph, 2022, doi:10.3390/ijerph191911954_

Round 1

Reviewer 1 Report

The authors displayed a thorough review with adequate methods and discussion. Some minor changes should be completed prior to publication as listed below.  

The introduction is well written and provides an appropriate background for the understanding PAP and CAs. However, not much is provided to explain the background for blood flow restriction, which is the main focus of the manuscript. Provide more background information regarding the mechanisms of action surrounding BFR to better understand the performance benefits proposed.

Explain and reference the controversial results mentioned. 

The conclusion section refers to this as a study rather than a review. This may be confusing to the reader and imply that novel data is being discussed. This should be changed for accuracy and transparency.

Author Response

    Relevant content regarding the conception, applications, and advantages of BFR training has been added in the third paragraph of the introduction. And because more details on physiological mechanisms are presented in the discussion, we didn't elaborate it in introduction. Currently, more researches intend to examine BFR training's effect on muscle hypertrophy and strength, however, less attention is paid to the muscle activation led by restricted blood flow. And that may support BFR training as an effective strategy to elicit PAP, furthermore, the lower loads requirement might make it a safer and more flexible condition activity compared to traditional heavy load training.

    We have altered the expression "study" to the expression "review" and we fully agree that it's important to clearly define the types of research for readers.

Reviewer 2 Report

The topic is relevant to the area of sport, training, and rehabilitation. Thus, a systematic literature review to establish evidence of its applicability is important.

The authors found few studies with different analysis variables and audiences with different age groups that point to a promising effect of BRF-ACs, however, the results cannot establish evidence, but rather point to promising results.

Especially in the discussion, the authors try to show evidence of the promising effects of BR-ACs, but the limitations pointed out indicate that more studies are needed to establish the type of training in PAP.

As a suggestion, in the discussion, the authors should clearly point out the weak points of each study and bring more elements that strengthen the need for new studies.

Based on their experience, the authors can indicate more specific suggestions for future research on methodological procedures: such as sample size, standardization of subjects (in terms of age, sex, time and type of training, diet, health history, race, etc.), analysis variables, types of equipment used.

It does not seem pertinent to me to include in the conclusion (article and abstract) about BFR requiring lower training loads and then powers to benefit untrained individuals and help save energy and reduce the risks of overweight and fatigue. As far as I understand, this was a discussion and not the results of the selected articles.

As a suggestion: Withdraw from the conclusion the statement: Our results provide qualitative evidence to support the advantage of BFR-CAs in inducing PAP. Replace with: The results point to the advantages of BFR-CAs in inducing PAP.

Authors should note lines 24, 25, and 96. There appear to be errors in the writing.

Author Response

    We have added some details and suggestions on rest intervals (line 185 ~ line 192), individual responses to CAs (line 193 ~ line 199), recruitment on further subjects (line 275~line 279), BFR conditions (section" Equipment tips for BFR-CAs") according to the included studies. For rest intervals, multiple rest intervals are suggested based on window time; for individual responses to CAs, we believe the variability among subjects should be taken seriously and suggest checking the individual response in advance; for further recruitment of subjects, we suggest examining the BFR-CAs effect on minor subjects and older subjects; for BFR conditions, pressure modality is suggested to construct based on subjects' limb characteristics and AOP.   

    We agree that more related researches require to be further carried out to support the original conclusion and we've altered the presentation in the conclusion section.  

Round 2

Reviewer 2 Report

The authors made the changes as requested, which contributed to the quality of the article.